# Curcumin as a Stabilizer of Macrophage Polarization during *Plasmodium* Infection

**DOI:** 10.3390/pharmaceutics15102505

**Published:** 2023-10-21

**Authors:** Maria Clara C. Cordeiro, Fernanda D. Tomé, Felipe S. Arruda, Simone Gonçalves da Fonseca, Patrícia R. A. Nagib, Mara R. N. Celes

**Affiliations:** 1Department of Bioscience and Technology, Institute of Tropical Pathology and Public Health, Federal University of Goiás, Goiânia 74605-050, GO, Brazil; mariac.coelho07@gmail.com (M.C.C.C.); fernandadiast@gmail.com (F.D.T.); arrudaf@ymail.com (F.S.A.); sfonseca@ufg.br (S.G.d.F.); 2Department of Microbiology, Immunology and Parasitology, Biological Science Institute, Federal University of Juiz de Fora, Juiz de Fora 36036-900, MG, Brazil; patricia.nagib@ufjf.br

**Keywords:** RAW 264.7 cells, curcumin, malaria, iNOS, arginase, phagocytosis

## Abstract

Malaria is a parasitic infection responsible for high morbidity and mortality rates worldwide. During the disease, phagocytosis of infected red blood cells by the macrophages induces the production of reactive oxygen (ROS) and nitrogen species (RNS), culminating in parasite death. Curcumin (CUR) is a bioactive compound that has been demonstrated to reduce the production of pro-inflammatory cytokines and chemokines produced by macrophages but to reduce parasitemia in infected mice. Hence, the main purpose of this study is to investigate whether curcumin may interfere with macrophage function and polarization after *Plasmodium berghei* infection in vitro. In our findings, non-polarized macrophage (M0), classically activated (M1), and alternatively activated (M2) phenotypes showed significantly increased phagocytosis of infected red blood cells (iRBCs) when compared to phagocytosis of uninfected red blood cells (RBCs) 3 h after infection. After 24 h, M1 macrophages exposed to RBCs + CUR showed greater elimination capacity when compared to macrophages exposed to iRBCs + CUR, suggesting the interference of curcumin with the microbicidal activity. Additionally, curcumin increased the phagocytic activity of macrophages when used in non-inflammatory conditions (M0) and reduced the inducible nitric oxide synthase (iNOS) and arginase activities in all macrophage phenotypes infected (M0, M1, and M2), suggesting interference in arginine availability by curcumin and balance promotion in macrophage polarization in neutral phenotype (M0). These results support the view of curcumin treatment in malaria as an adjuvant, promoting a balance between pro- and anti-inflammatory responses for a better clinical outcome.

## 1. Introduction

Malaria is a parasitic infection caused by *Plasmodium* species and transmitted to humans through the bite of an infected female *Anopheles* mosquito [1,2]. This disease is one of the main causes of morbidity and mortality around the world [3,4], and in 2021, 247 million cases and 619,000 deaths were registered globally [5]. Additionally, the attempts to reduce coronavirus transmission during the global pandemic declared in 2020 raised concerns in malaria-endemic countries for creating difficulties to use essential services such as seasonal malaria chemoprevention and insecticide-treated bed nets distribution, increasing the vulnerability of these populations [5,6].

The first asymptomatic stage of the disease takes place in the liver, and the immune system begins to respond by the action of dendritic cells, natural killer cells, and macrophages, which act as antigen-presenting cells and perform phagocytosis [6,7]. The second, or erythrocytic, phase highlights the importance of macrophages to perform phagocytosis of red blood cells infected with *Plasmodium* and produce reactive oxygen species (ROS) and nitrogen species (RNS), crucial actions for parasite death that allows the regulation of parasitemia [7,8].

During infection, the malarial antigens activate the macrophages and undergo several metabolic and functional changes, culminating in their physiological alterations and giving rise to different functional phenotypes. Depending on the activating stimuli in the microenvironment, macrophages can be classically (M1) activated and are responsible for pro-inflammatory responses such as phagocytosis and microbicidal activity, while the alternatively activated (M2) macrophages act in tissue remodeling and inflammation resolution [8]. This polarization process is dependent on the cytokines produced, the duration of antigen exposure, and the presence of growth factors, fatty acids, prostaglandins, and pathogen-derived molecules. These stimuli act as signaling cascades, such as the mTOR/PI3K/Akt pathway, which is described as the central regulator of the macrophage activation phenotype. Furthermore, changes in this microenvironmental stimulus can reverse the original polarization, characterizing the plasticity of macrophages [9,10,11,12,13].

In malaria, the equilibrium between these responses is essential for the best clinical outcome; insufficient pro-inflammatory factors can lead to uncontrolled growth of *Plasmodium*, while their excessive quantity can cause tissue damage, anemia, and the development of more severe conditions, such as cerebral malaria [12,13,14].

Some natural compounds can promote the production of different inflammatory factors, which stimulate or inhibit the inflammatory response. Bioactive compounds, such as curcumin, obtained from the dried rhizomes of Curcuma L. Longa, have pharmacological antitumor, antimicrobial, and antiangiogenic activities and can be easily incorporated into the diet without any adverse effects [15]. Approved as a “safe compound” by the US Food and Drug Administration (FDA), curcumin generally is well tolerated, even in high doses [16]. Moreover, curcumin derivatives exhibit a wide range of effective actions at the cellular level, including the regulation of transcription factor levels, such as NF-kappaB (NF-κB), the modulation of inflammatory cytokines expression, and the production/activation of enzymes involved in the progression of infections [15,17]. Experimental studies that were performed in mice infected with *Plasmodium* demonstrated that curcumin reduced parasitemia, increased animal survival [18], and promoted neuroprotection during cerebral malaria [19].

Curcumin attenuates the production of pro-inflammatory cytokines and chemokines by macrophages, interfering with some important cellular pathways [15,17], including the regulation of mTOR/PI3K/Akt involved in the macrophage activation phenotype [18,19,20]. Therefore, we hypothesized that curcumin could interfere with macrophage function, inducing changes and polarization after infection with *Plasmodium berghei* in vitro.

## 2. Materials and Methods

### 2.1. Reagents and Animals

The following reagents and cytokines were used: RPMI-164 medium (Cultilab, Waltham, MA, USA), mouse recombinant IFNγ and mouse recombinant IL-4 (Peprotech, Rocky Hill, CT, USA), *Escherichia coli* LPS and zymosan from Saccharomyces cerevisiae (Sigma-Aldrich, St. Louis, MO, USA), and curcumin (Farmácia Escola from the Federal University of Goiás, BRA).

Animal studies were previously approved by the Animal Ethics Committee of the Federal University of Juiz de Fora (protocol number 019/2018). Experiments were performed in 6–8-week-old male Balb/c mice, specific pathogen-free (SPF). The animals were housed in a temperature-controlled room (23–25 °C), under a 12-h light-dark cycle, with water and food ad libitum.

### 2.2. Cell Culture and Polarization Treatments

The RAW 264.7 macrophage cell line (Rio de Janeiro Bank Cells, Duque de Caxias, Brazil) was cultured in the RPMI-1640 medium supplemented with 10% fetal bovine serum, 2 mM/mL Glutamine, 7.5% 12 mM bicarbonate solution, and 1% antibiotic solution composed of penicillin and streptomycin, at 37 °C with 5% CO_2_ in a humidified incubator. After 14 days, 3 × 10^5^ cells/well were dispensed into each well of 24-well plates and incubated at 37 °C in 5% CO_2_ for 4 h. Into each well, the following treatment for macrophage polarization was added: LPS (3 μg/mL) + IFNγ (15 ng/mL) for M1 phenotype; IL-4 (60 ng/mL) for M2 phenotype; or only the RPMI-1640 medium supplemented for M0 macrophages (control group). These cells were incubated for 24 h. The confirmation of cell polarization (M1 or M2 phenotypes) is detailed in Appendix A (Figure A1).

### 2.3. Curcumin Concentrations

To determine the best concentration of curcumin to be used in the experiments, 1 × 10^5^ cells/well were plated, polarized, and treated with filtered curcumin at 1, 5, or 10 μM, dissolved in 0.5% of dimethyl sulfoxide (DMSO) plus the RPMI-1640 medium. After 3 or 24 h, supernatant samples were collected for the NO production assay and cell samples for the arginase activity analysis (Appendix A, Figure A2 and Figure A3). The MTT assay was also performed using 1 × 10^4^ cells/well, plated following the same steps described. The results showed that curcumin filtered at 5 μM is able to inhibit the iNOS enzyme in M1 macrophages without significant stress to cells (Appendix A, Figure A4) and thus was the concentration chosen for the next steps.

### 2.4. Cell Infection with Plasmodium berghei NK65 and Treatment with Curcumin

Red blood cells and red blood cells infected with *Plasmodium berghei* NK65 were used in the next steps. After differentiation, macrophages M0, M1, and M2 (3 × 10^5^ cells/well) were cultivated in 24-well plates containing a sterilized coverslip (13 mm diameter), and each macrophage group (plated at 5 × 10^5^ cells/well) was incubated with the following treatment: zymosan A, a polysaccharide from *Saccharomyces cerevisiae* (10 particles/cell) for phagocytosis control, non-infected red blood cells (10 RBCs/cell) obtained from healthy BALB/c mice to control phagocytosis, infected red blood cells (10 iRBCs/cell) from *P. berghei*-infected mice on the 14th day of infection, curcumin (5 μM), zymosan plus curcumin, RBC plus curcumin, or iRBC plus curcumin. All of the groups were incubated for 3 or 24 h at 37 °C with 5% CO_2_.

### 2.5. Phagocytosis and Microbicidal Activity

To investigate the phagocytic and microbicidal capacity, the supernatants were collected after 3 and 24 h, and the coverslips were washed twice with PBS, removed, fixed in paraformaldehyde 4%, and stained with Romanowsky stain. Using light optical microscopy (40×), we counted 200 macrophages per well and the number of particles (zymosan, RBC, or iRBC) internalized by each of them at 3 and 24 h. The mean number of particles per cell was determined according to the following calculation: the number of phagocytized particles divided by the total infected macrophages. The infection index was calculated according to the following formula: the average of the phagocyted particles multiplied by the percentage (%) of cells containing at least one phagocyted particle.

### 2.6. NO Production Assay

The nitrate concentrations in the supernatant collected from the cultured medium were measured using a Griess reaction. In 96-well plates, a sample or nitrite was pipetted in duplicate with the Griess reagent (1% sulfanilamide and 0.1% naphthyl ethylenediamine) under acidic (phosphoric acid) conditions and incubated in the dark at room temperature for 10 min. The values were read on a spectrophotometer at the length of a 550 nm wave, and the calculations of the nitrite concentrations in the samples were carried out based on the known values of the standard curve.

### 2.7. Arginase Activity Assay

The activity of the enzyme l-arginase generates some products, among them urea. For the urea dosage, the macrophages were collected from the bottoms of the plates using PBS. Each sample was added to 1 mM MnCl_2_ before incubation in a thermoblock system (used for rapid heating and rotation of the sample) at 56 °C for 10 min at 400 rpm. The samples were transferred to ice and were added to L-arginine (5 M). One group was incubated on ice for 1 h to measure the basal activity, while the other was incubated for 1 h in a thermoblock at 37 °C (400 rpm). Then, an acid solution (1 H_2_SO_4_:3 H_3_PO_4_:7 H_2_O) was added and α-isonitrosopropiophenone for both samples and curves, followed by incubation for 45 min at 97 °C (600 rpm), followed by another incubation period in the dark at 35 °C (10 min and 600 rpm). The values were read on a spectrophotometer at the length of a 550 nm wavelength.

### 2.8. Statistical Analyses

The data obtained were analyzed using the GraphPad Prism 8 statistical program (Graph Pad Software Inc., San Diego, CA, USA). Student’s *t*-test was used to compare two normally distributed variables. Multiple comparisons were made by analysis of variance (one-way or two-way ANOVA), followed by the Tukey post-hoc test. The significance level chosen was 5%, and data are presented as mean + standard deviation (SD).

## 3. Results

### 3.1. Pro-Inflammatory Macrophages Increased Phagocytosis and Microbicidal Activity of iRBCs

After in vitro polarization, the macrophages were incubated with red blood cells infected with *Plasmodium berghei* NK65 (iRBCs) and subjected to an analysis of the phagocytic and microbicidal activities. Macrophage phenotypes M0, M1, and M2 showed significantly increased phagocytosis of iRBCs when compared to phagocytosis of uninfected red blood cells (RBCs). Three hours after incubation with iRBCs, the mean number of phagocytosed particles (Figure 1A,C,E) and the infection index of the macrophages (phagocytic cells) (Figure 1B,D,F) increased four-fold for the pro-inflammatory M1 phenotype (Figure 1C,D). When treated with CUR, macrophage phenotypes M0, M1, and M2 presented a similar performance.

To understand whether curcumin influenced the function of polarized and infected macrophages, the phagocytic activity was evaluated three hours after the curcumin treatment. Interestingly, the treatment of polarized macrophages (M1 or M2) with curcumin did not alter the phagocytic activity of the macrophages incubated with iRBCS or RBCs. However, in polarized M1 or M2 macrophages incubated with zymosan (Figure 1), curcumin significantly increased the phagocytosis index, with an increase in the average number of phagocytosed particles per cell three hours after infection and treatment.

### 3.2. Curcumin Increased Phagocytosis in Non-Infected and Polarized Macrophages

The microbicidal activity was evaluated 24 h after infection and incubation with curcumin. The average number of particles inside non-polarized macrophages (M0) was similar in all studied groups 24 h after infection (iRBCs) and treatment with CUR (iRBCs + CUR) (Figure 2A). However, the M0 macrophages incubated with RBCs plus curcumin had a lower macrophage infection index (Figure 2B) when compared to M0 cells incubated only with red blood cells, indicating that curcumin can improve the ability of the M0 macrophages to eliminate RBCs. The M1 macrophages incubated with red blood cells plus curcumin (RBC + CUR) revealed an average number of particles (Figure 2C) and a higher macrophage infection index (Figure 2D) when compared to M1 cells incubated with infected red blood cells (iRBC + CUR). The average number of particles (Figure 2E) and macrophage infection index (Figure 2F) were similar in all studied groups 24 h after infection and treatment with CUR.

### 3.3. Curcumin Reduced the Microbicidal Activity of the Pro-Inflammatory Macrophages

The killing ability of the macrophages was evaluated 24 h after infection with infected or uninfected RBCs. The mean number of phagocytosed particles was significantly reduced compared to the results observed three hours after infection (Figure 2), demonstrating that the previously phagocytosed particles were eliminated by the microbicidal activity of the macrophages (Figure 3). This microbicidal activity was observed only in the cells infected with *Plasmodium*, with the percentage of activity higher in the M1 phenotype. Furthermore, infected M1 cells, when incubated with curcumin, showed a decreased number of cells with microbicidal activity compared to cells without curcumin treatment (Figure 3), demonstrating that in this case, curcumin was able to interfere with the elimination of particles previously internalized by the M1 phenotype.

### 3.4. iNOS Enzyme Had No Participation in Macrophage Microbicidal Activity after Plasmodium Infection and Curcumin Treatment

Previous studies by our group (summarized in Appendix A) demonstrated that curcumin was able to inhibit the iNOS enzyme in M1 macrophages 24 h after being exposed to uninfected polarized macrophages. To clarify whether this microbicidal activity occurred in the absence of the iNOS enzyme, an NO production assay was performed 24 h after infection and treatment with curcumin.

iNOS activity is considered a hallmark marker of the M1 macrophage, and as expected, nitrite was not detected in the non-polarized macrophages (M0). Additionally, the M2 macrophages showed decreased levels of nitrite (Figure 4); the group that received iRBCs was an exception. We also observed reduced levels of nitrite in the M1 macrophages (Figure 4B), despite the increase in phagocytosis and microbicidal activity observed previously, which demonstrates the lack of participation of the iNOS enzyme in the microbicidal activity of the macrophages after 24 h of *Plasmodium berghei* infection and treatment with curcumin.

### 3.5. Curcumin Decreased Arginase Activity in Infected and Non-Infected Macrophages

As arginase is a marker of anti-inflammatory macrophages (M2) in cell culture, the arginase activity assay was performed to elucidate whether curcumin influenced urea formation and nitrite formation, which were previously observed in polarized and infected macrophages.

The results showed that curcumin interfered with urea formation. All of the macrophage phenotypes (M0, M1, and M2) demonstrated reduced urea levels 24 h after infection and curcumin treatment (Figure 5), indicating a decrease in arginase enzyme activity. Furthermore, the production of urea by macrophages that received RBCs (Figure 5), even after treatment with curcumin, demonstrates that these RBCs act as an anti-inflammatory stimulus when added to a macrophage cell culture.

### 3.6. Curcumin Stabilized Macrophage Polarization during Plasmodium Infection

Several studies have shown that macrophage polarization is not definitive and can be altered according to the stimuli received in the microenvironment. Considering this, we reanalyzed the phagocytic activity shown in Figure 1 and observed that the M1 and M2 cells treated with curcumin showed similar behavior. Statistical analyses showed no difference between these groups, suggesting that curcumin can avoid the most extreme polarization phenotypes, acting as a stabilizer of macrophage polarization during *Plasmodium* infection (Figure 6).

## 4. Discussion

This investigation sought to understand the in vitro effects of curcumin in the function of polarized murine macrophages exposed to red blood cells infected with *Plasmodium berghei* NK65 or uninfected. Firstly, the phagocytic model was reproduced in vitro and worked successfully once all of the macrophage phenotypes were able to phagocytize zymosan, which is a model of classical phagocytosis showing interaction with immunity cells [20]. Likewise, we demonstrated that iRBCs behaved like PAMPs [21], and the M0, M1, and M2 macrophages were able to recognize and phagocytose these iRBCs [22].

In this study, we demonstrated phagocytosis occurring at higher levels when macrophages were in contact with red blood cells infected by *Plasmodium berghei*. We observed a pattern of a higher average of phagocyted particles and an increased percentage of phagocytic cells when these were in contact with iRBCs instead of RBCs, as observed by Shear and colleagues in 1979 [23]. Additionally, the M1 phenotype, known as pro-inflammatory with high phagocytic and microbicidal activities [12], displayed higher levels of phagocytosis than the M0 and M2 phenotypes. Correlating with these results, microbicidal activity was clear in all of the macrophage phenotypes infected, as well as phagocytic activity [11]; the average number of phagocyted particles inside the macrophages decreased after 24 h when compared to 3 h after the infection. All of these findings demonstrate that the macrophages not only recognized iRBCs but also ingested and destroyed these iRBCs, with significant phagocytic and microbicidal functions. The presence of *Plasmodium berghei* inside erythrocytes could potentiate PAMP’s activation and related pathways, increasing cytokine production and promoting phagocytosis and digestion. In this context, there has been an increase in MHC class II and co-stimulatory molecules and an increase in cytokine production in infected macrophages after internalization of *Plasmodium* in RBC, involving TLR4, TLR9, and MyD88, and signaling via NF-κB [24,25].

To evaluate whether curcumin could interfere with phagocytic and microbicidal activities, the previously polarized macrophages were analyzed at 3 and 24 h after treatment with curcumin. We observed that the curcumin treatment did not change these activities in the environment in vitro with *Plasmodium* infection but increased phagocytosis in the non-infected and polarized macrophages. Previous in vitro and in vivo studies suggest that curcumin should be used as adjunctive therapy [26,27,28,29]. Although curcumin did not interfere positively or negatively in phagocytic and microbicidal activities when macrophages received iRBCs, our results demonstrate that curcumin significantly increased phagocytosis in polarized M1 or M2 phenotype macrophages three hours after receiving zymosan. These data agree with previous results that observed up-regulation in the phagocytic activity of RAW 264.7 macrophages after curcumin treatment under non-inflammatory conditions [27,30]. In our findings, the cells were previously stimulated with LPS, as the M1 macrophages in this study were differentiated in the presence of LPS. Thus, our results confirm that treatment with curcumin enhances the phagocytic activity of macrophages, independently of the activation received (M1 or M2), when compared with non-treated macrophages.

The M1 and M2 macrophages infected and treated with curcumin displayed similar behavior to non-polarized macrophages infected without curcumin treatment. Considering that macrophages can reverse the original polarization according to changes in the microenvironmental stimuli, it is possible that curcumin treatment could affect the polarization process; the attenuation of pro-inflammatory cytokines and chemokines produced by macrophages and influenced by this compound [14,15,31] can change the pro-inflammatory responses. This suggests that curcumin can avoid the more extreme phenotypes of M1 and M2 polarization, and these cells can behave like unpolarized and infected macrophages, still carrying out phagocytic and microbicidal activities but with the polarization stabilized in a neutral phenotype during *Plasmodium* infection. This action could favor the use of curcumin as an adjuvant in the treatment of malaria, since a balance between pro- and anti-inflammatory responses is essential for the best clinical outcome in this disease [32].

During phagocytic and microbicidal activities, reactive nitrogen species (RNS) are released by macrophages, like nitric oxide (NO) through the enzyme nitric oxide synthase (iNOS) [33]. This process is dependent on the availability of the amino acid arginine, which is converted into NO and citrulline in M1 cells [34]. Previous results showed that curcumin treatment inhibits iNOS enzyme production by M1 macrophages [19,27,30]. Thus, our results confirm that macrophages perform phagocytosis and microbicidal activity in the absence of the iNOS enzyme through the low nitrite values detected in all of the macrophage phenotypes, indicating that the iNOS enzyme did not participate in the microbicidal activity of the macrophages and suggesting that other RNS or ROS produced by these cells could destroy the phagocyted particles. A similar analysis occurred in a study of the killing of *Leishmania braziliensis* by monocytes, which concluded that NO alone is not sufficient to control the infection in patients with Cutaneous Leishmaniasis, while the production of ROS is involved in parasite killing [35].

After looking into the iNOS enzyme, we analyzed the activity of the arginase enzyme, since both make use of arginine as a substrate to produce their compounds [36]. Arginase converts arginine into ornithine and urea in the M2 cells [37]. In this study, we observed that the treatment of macrophages with curcumin after polarization reduced the levels of urea by all macrophage phenotypes, indicating a decrease in the activity of the arginase enzyme. Furthermore, a reduction in serum urea was observed in mice that received nanometric curcumin, with a protective effect on the kidneys of infected animals that is similar to chloroquine treatment [26].

Outside the malaria context, renal toxicity and carcinogenesis studies have also observed reduced arginase activity after curcumin treatment [34,38]. Unexpectedly, our results leave us without an explanation for how curcumin reduces the activity of both enzymes iNOS and arginase in the M0 or polarized M1 and M2 macrophages. Since both enzymes depend on the arginine supply, curcumin could interfere with arginine availability by hampering the enzyme’s binding site or by altering important cofactors in enzymatic reactions [38,39]. Although the mechanism is unclear, curcumin may have prevented the availability of arginine to macrophages, inhibiting the production of the markers of the M1 and M2 phenotypes (NO and urea, respectively), which may be related to the possible ability of curcumin to balance polarization, avoiding the two extreme phenotypes and consequently not reducing arginase activity in a significant way [39].

## 5. Conclusions

In summary, this study demonstrates that the phagocytic and microbicidal activities of macrophages, when in contact with *Plasmodium berghei*-infected red blood cells, are high, mainly in the pro-inflammatory M1 phenotype. In addition, curcumin stimulated phagocytic and microbicidal activity when used under non-infectious conditions, bringing new perspectives to the use of curcumin in maintaining phagocytic activity in circumstances of cell turnover and phagocytosis of altered organism particles. Our findings provide insight into curcumin treatment in malaria as an adjuvant, promoting equilibrium between pro- and anti-inflammatory responses for a better clinical outcome, once curcumin was able to reduce the NO and urea production in all of the macrophage phenotypes. More experiments should elucidate the mechanism involved in this process, clarifying its possible relationship with the arginine blockade and balance promotion in macrophage polarization in the M0 neutral phenotype.

## Figures and Tables

**Figure 1 pharmaceutics-15-02505-f001:**
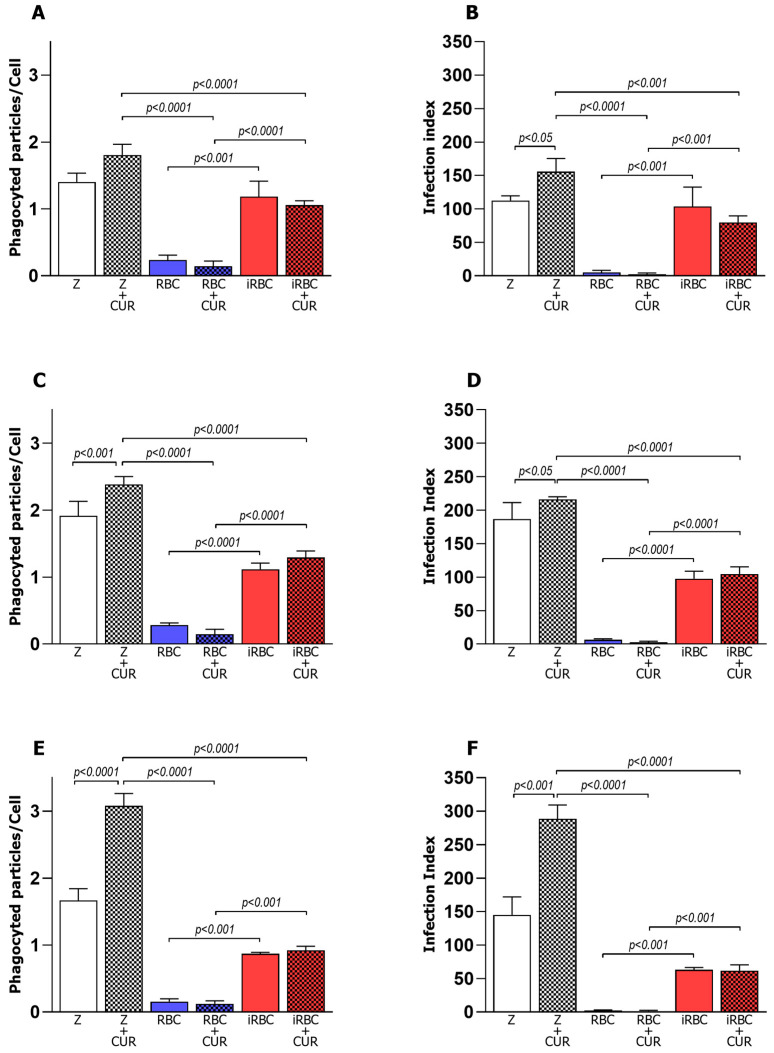
Macrophage phagocytic activity three hours after *Plasmodium berghei* NK65 infection and curcumin treatment. Average number of particles per cell by M0 (**A**), M1 (**C**), and M2 (**E**) macrophages and infection index of M0 (**B**), M1 (**D**), and M2 (**F**). CUR = curcumin, Z = zymosan, RBC = red blood cells, iRBC = red blood cells infected with *Plasmodium berghei*. Data were calculated from N = 3 and are presented as mean ± standard deviation. *p*-values are indicated and are considered statistically significant when *p* < 0.05.

**Figure 2 pharmaceutics-15-02505-f002:**
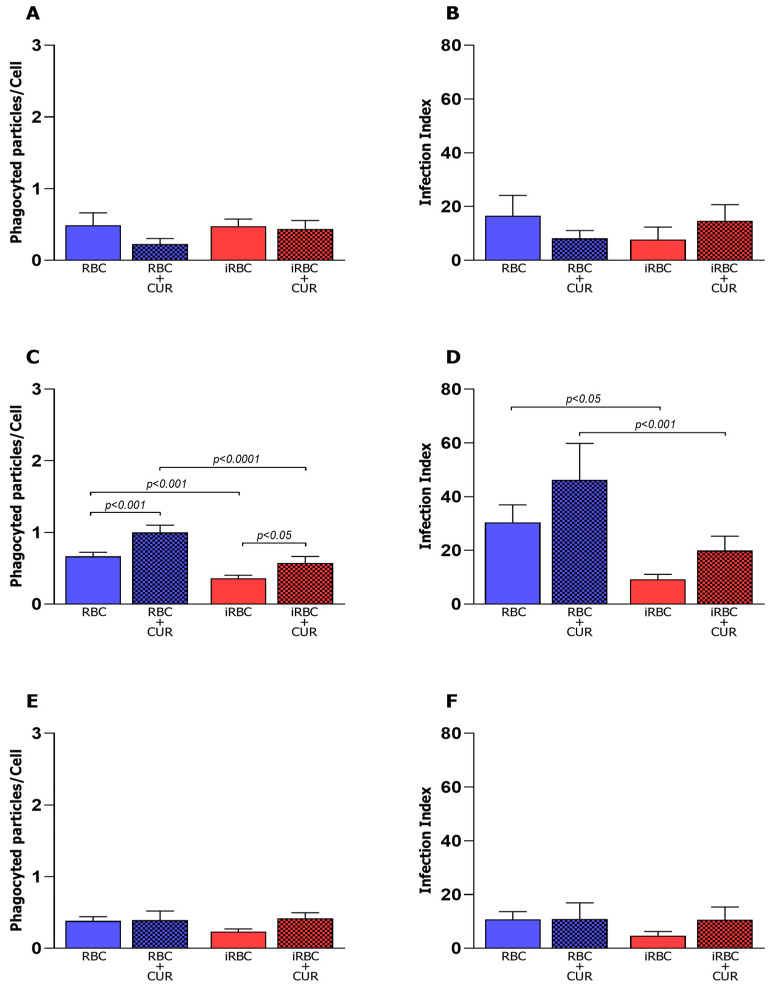
Macrophage phagocytic activity 24 h after *Plasmodium berghei* NK65 infection and curcumin treatment. Average number of particles per cell by M0 (**A**), M1 (**C**), and M2 (**E**) macrophages and infection index of M0 (**B**), M1 (**D**), and M2 (**F**). CUR = curcumin, RBC = red blood cells, iRBC = red blood cells infected with *Plasmodium berghei*. Data were calculated from N = 3 and are presented as mean ± standard deviation. *p*-values are indicated and are considered statistically significant when *p* < 0.05.

**Figure 3 pharmaceutics-15-02505-f003:**
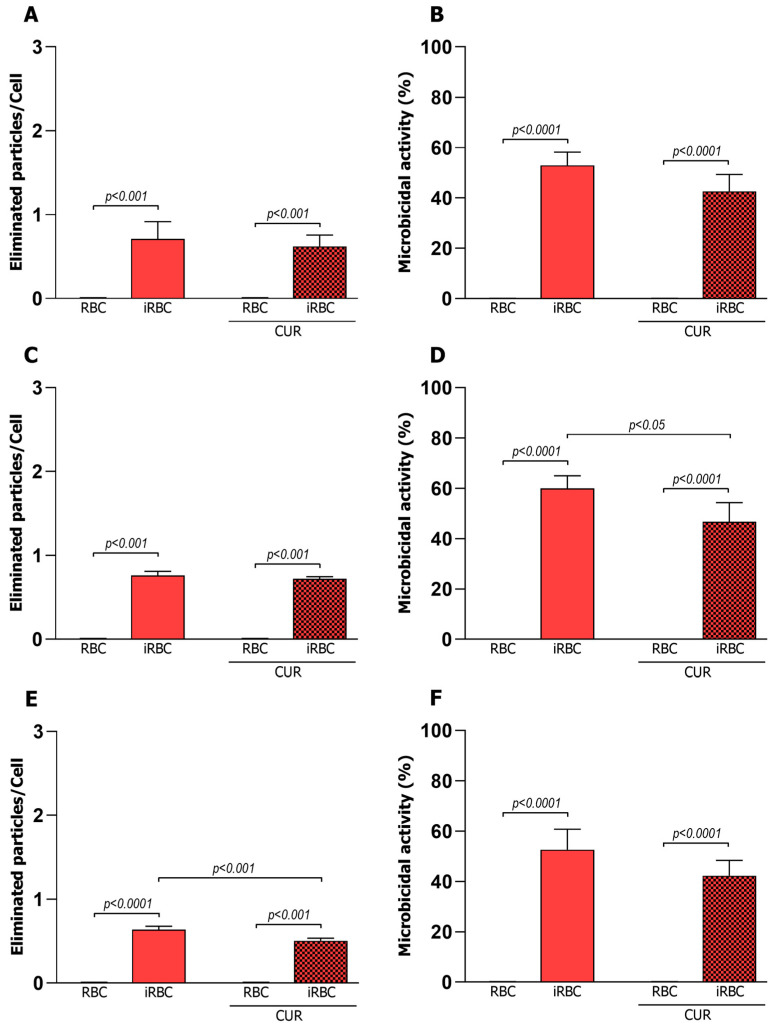
Microbicidal activity of macrophages assessed 24 h after *Plasmodium berghei* NK65 infection and curcumin treatment. Average of eliminated particles per cell by M0 (**A**), M1 (**C**), and M2 (**E**) macrophages and percentages of M0 (**B**), M1 (**D**), and M2 (**F**) macrophages that displayed microbicidal activity. CUR = curcumin, RBC = red blood cells, iRBC = red blood cells infected with *Plasmodium berghei*. Data were calculated from N = 3 and are presented as mean ± standard deviation. *p*-values are indicated and are considered statistically significant when *p* < 0.05.

**Figure 4 pharmaceutics-15-02505-f004:**
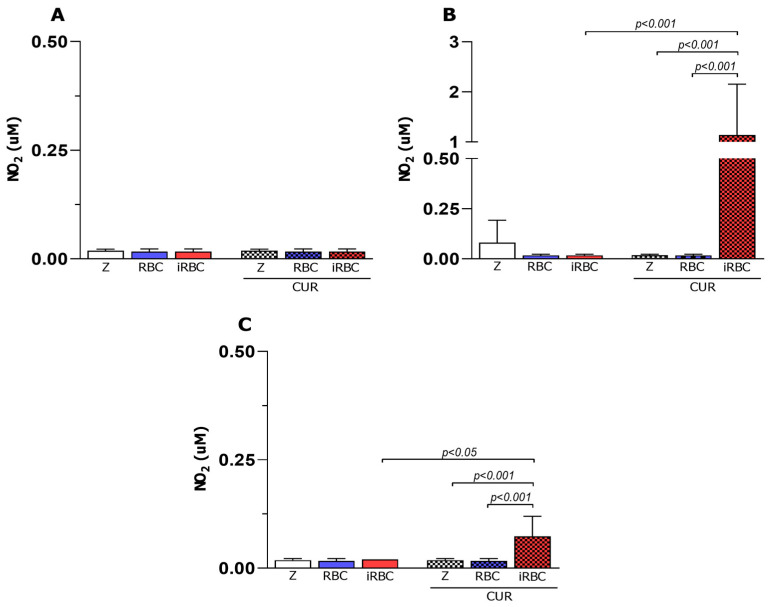
Nitrite production 24 h after *Plasmodium berghei* NK65 infection and curcumin treatment. Concentrations (μM) detected in M0 (**A**), M1 (**B**), and M2 (**C**) macrophages were calculated from N = 6 (Z and CUR groups) and N = 3 (RBC, iRBC, RBC + CUR, and iRBC + CUR). Z = zymosan, CUR = curcumin, RBC = red blood cells, iRBC = red blood cells infected with *Plasmodium berghei*. Data are presented as mean ± standard deviation. *p*-values are indicated and are considered statistically significant when *p* < 0.05.

**Figure 5 pharmaceutics-15-02505-f005:**
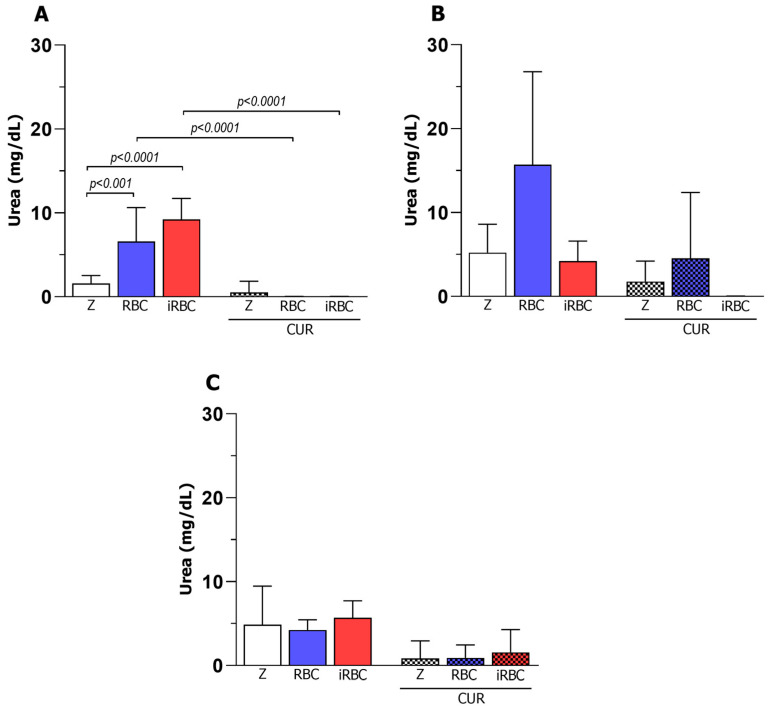
Urea production 24 h after *Plasmodium berghei* NK65 infection and curcumin treatment. Concentrations (mg/dL) detected in M0 (**A**), M1 (**B**), and M2 (**C**) macrophages were calculated from N = 6 (Z and CUR groups) and N = 3 (RBC, iRBC, RBC + CUR, and iRBC + CUR). Z = zymosan, CUR = curcumin, RBC = red blood cells, iRBC = red blood cells infected with *Plasmodium berghei*. Data are presented as mean ± standard deviation. *p*-values are indicated and are considered statistically significant when *p* < 0.05.

**Figure 6 pharmaceutics-15-02505-f006:**
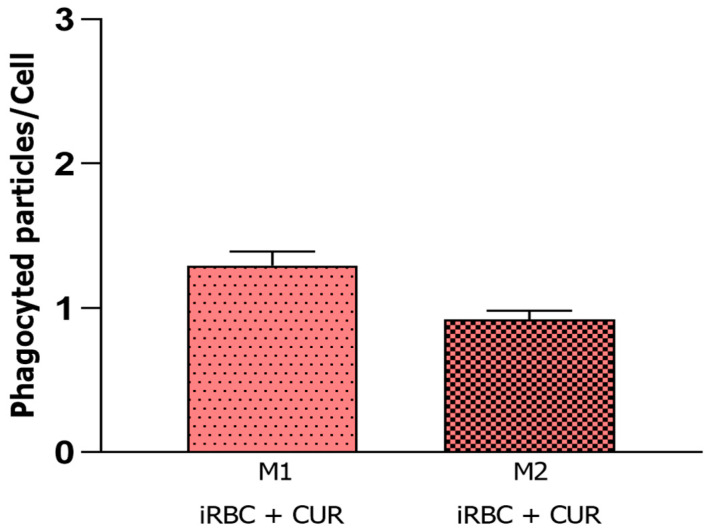
Macrophage phagocytic activity 3 h after *Plasmodium berghei* NK65 infection. Average phagocyted particles per cell by M1 and M2 macrophages after curcumin treatment. CUR = curcumin; iRBC = red blood cells infected with *Plasmodium berghei*. Data were calculated from N = 3 and are presented as mean ± standard deviation. *p*-values are indicated and are considered statistically significant when *p* < 0.05.

## Data Availability

All data are contained within the article.

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
