# Peer review of "Curcumin as a Stabilizer of Macrophage Polarization during Plasmodium Infection"

_pharmaceutics, 2023, doi:10.3390/pharmaceutics15102505_

Round 1

Reviewer 1 Report

The manuscript by Maria C. C. Cordeiro et al. deals with the investigation of the effect of curcumin on polarization and function of macrophages after infection with Plasmodium berghei in vitro.

The manuscript is written well and deals with a timely relevant topic. However, there are some issues that needs to be revised/clarified before publication in ‘Pharmaceuticals’:

Comments:

1. The hypothesis of this work needs to be worked out better.

2. Why the authors decided to use a murine macrophage cell line (RAW 264.7), viral infected producing viral particles (Hartley JW, et al., (2008) Expression of infectious murine leukemia viruses by RAW 264.7 cells, a potential complication for studies with a widely used mouse macrophage cell line. Retrovirology 5:1). Therefore, it is hardly to believe, that these cells represent M0 macrophages, although cultured under M0 conditions. A more suitable system would be a human macrophage cell line or even better in vitro generated macrophages from peripheral blood monocytes.

3. The authors suggest that curcumin stabilizes macrophage polarisation. This suggestion arose from one experiment depicted in Figure 6.  As the central regulator of the macrophage polarization is the mTOR/PI3K/Akt pathway, the authors should provide evidence for their suggestion by analysing this pathway and/or by providing additional support of this data interpretation. Otherwise the evidence is missing. Since the manuscript is entitled “Curcumin as a stabilizer of macrophage polarization during Plasmodium infection”, this point should be worked out intensively.

4. The quality of figure has to be improved. The font size must be adjusted accordingly. As the authors compare M0, M1 and M2 polarized macrophages, these data should be presented in one graph that one can compare the effects directly. 

Author Response

Dear Reviewer,

All comments and point-by-point responses are provided in the PDF document.

Mara Celes

Reviewer 2 Report

The article “Curcumin as a stabilizer of macrophage polarization during Plasmodium infection” by Cordeiro et al. is an experimental study examining how curcumin could modulate the function of macrophages and their polarization after infection with Plasmodium berghei.

However, the article has many shortcomings as follows:

1.      The title is not correctly written in the all-caps title style (please, see the correction in the title, stipulated in RED).

2.      For the keywords: The authors did not adhere to MeSH for choosing keywords and have several imperfections in keyword formulations. There are keywords that are already in the title and therefore should NOT have been listed as keywords (eg., macrophage polarization; curcumin).

3.      The abstract is not written rigorously and credible, so it should be reformulated in the light of experiments, controllers, results, and conclusions obtained.

4.      The editing was done in a hurry, often with carelessness throughout the text, for example:

-        unjustified introduction of an abbreviation in the abstract, for example (MO), when it is subsequently NOT used in the abstract. In general, it is best to avoid using abbreviations and acronyms in the abstract unless the abbreviation/acronym is used multiple times in the abstract. Pharmaceutics may not allow the usage of abbreviations in the abstract.

-        the number of cells is written incorrectly, for example, see:

lines 370: 3x105 cells/well

or 378: 1x105 cells/well.

And the examples can continue…

5.      The postulation of some statements from the introduction based on the citation of only one article does not correspond to the scientific rigor related to Pharmaceutics.

6.      A better representation of the experiments and the work is needed for the reader, better explaining the experimental control methods to demonstrate the results!!!

7.      The concise way how the experiments were carried out is missing for the reader to understand this article.

8.      The problem of the infection of macrophages with Plasmodium is treated superficially, it is neither credible nor properly justified from a scientific point of view.

9.      Each stage of the experiment must be rewritten and discussed objectively, correctly highlighting the control group.

10.   More comprehensive arguments and explanations of the results presented in all Figures would be most welcome, and the figures should be redesigned in a color-blind friendly palette.

11.   The figures do not suggest anything to the reader, being black and white and with very similar patterns. The Legend of each figure is written in the same size as the text and is confused with it.

12.   Figures are NOT correctly sized.

13.   The whole article should be improved in Results and Discussions.

More comparative arguments regarding the actual results obtained are missing. Discussions should better interpret the meaning of the results and explain why they matter.

14.   This article has NO conclusions!!!  Conclusions must be separated from Discussions.

15.   References should be double checked and improved with a digital object identifier (DOI), in MDPI and Pharmaceutics style. For example, http://dx.doi.org/10.9993/ajae/aaq93, which will take the reader directly to the information page for the article.   

16.   A “List of Abbreviations” must be completed and reviewed carefully and may be better presented in a table format at the end of the article.

17.   Academic editing of English, grammar, and style is required. The article is written as a "story"!

18.   I generally recommend a major revision.

I believe that after this revision provided by the authors on the issues suggested to be corrected and improved, it will provide useful and credible information for all readers, especially researchers and it is up to the Academic Editor to decide on its publication.

Thank you very much!

Academic editing of English, grammar, and style is required.

Author Response

Dear Reviewer,

All comments and responses are provided in the PDF document.

Mara Celes

Reviewer 3 Report

The manuscript of Cordeiro and colleagues investigates the contribution of curcumin on macrophages activity when exposed to plasmodium infected red blood cells. The authors demonstrated that curcumin does not affect the phenotype of macrophages but increased its phagocytic and microbicidal ability.

Although the topic is of interest, the results are convincing and well discussed, in the present form the manuscript needs to be revised.

Concerns

To state that curcumin does not affect macrophages polarization, investigation of iNOS/nitrates levels and arginase activity is not sufficient. The authors need to perform analysis of M1 or M2 markers (as reported in appendix A) in presence of curcumin. What is clear is that curcumin does not reverse the cytokine induced phenotype of macrophages. That need to be clarified throughout the manuscript.

Paragraph on curcumin concentration in material and methods needs to be revised. The statement: “Results have shown curcumin filtered at 5 μM was able to inhibit the iNOS enzyme in M1 macrophages without significant stress to cells (Appendix A, Figure A4), being the concentration chosen for the next steps.” Is not clear, please clarify better the significance of “stress to cells”. Further, as it is a result, this part should be reported in the “Results” section.

Please clarify better as the authors measured the phagocytic activity and the microbicidal activity.

In Material and Methods, the authors described the PC and PI, while results are reported as phagocyted particles/cell and phagocytic cells (%). Please change accordingly.

Findings in figure 1, as reported, are not clear: to compare the activity of M0 vs M1 and M2 alone or exposed to curcumin on zymosan, RBC and iRBC, the data should be reported in the same graph. Moreover, statistical analysis needs to be revised for all figures. Clarify the significance of “statistical bars” in the graphs in figure legends.

3.1 and 3.2 paragraphs refer to figure 1, for clarity’s sake, move figure 1 after the 3.2 paragraph, or make a single paragraph.

As for figure 1, panels of figure 2 and 3 should be associated in a single graph. Further, in figure legends of figure 2, 3, 4, 5 and 6, remove Z=zymosan as activity of macrophages on the stimulus has not reported.

In figure legend of figure 6 also remove RBC as macrophages are not exposed to these cells, and statistical analysis, as it is not reported.

Quality of English language is fine

Author Response

(The authors gave the same response as above.)

Reviewer 4 Report

The manuscript submitted by Cordeiro et al. reports on the studies investigating the effects of curcumin on macrophage activity in Plasmodium berghei infected cells in vitro. Phagocytic cell activity increased and iNOS and arginase activity was reduced as a result of curcumin treatment.

Comments: The use of bioactive compounds for direct treatment or  augmentation for treatment regimens for malaria is timely, due to the development of resistance to existing standard therapies. Several aspects of the study need clarification and the manuscript needs to be corrected for English syntax and grammar. Genus and species names should also be italicized. Exponents in the methods section need to corrected. Additional background information regarding curcumin is needed in the introduction.

Abstract

Line 14:  “…infected red blood cells by (the) macrophages…” delete “the”.

Line 16: “…a bioactive compound was demonstrated to reduce…”.

Introduction

Line 28: Plasmodium species

Lines 30-31: provide updated WHO statistics for malaria

Lines 36-42: revise and correct the grammar in this section.

NK cells are not antigen presenting cells and do not phagocytose

delete “second” and write “in the erythrocytic stage”

delete ”should be highlighted the importance of…”

Line 43: “During infection, malaria antigens activate…”

Line 45: “Depending on the …”

In lines 46-71, there are many more corrections like those indicated above.

Line 64: What Plasmodium species?

Line 67: Specify “some”. What cellular pathways are affected by curcumin?

Materials and Methods

Lines 74 and 81: Is this RpMI-1640?

Line 91: What was curcumin dissolved in (water, alcohol, DMSO)?

Line 92: Why were the times 3- or 24 hours chosen?

Lines 93-93: Rephrase “It was also performed MTT assay…”

Lines 95-97: move to results section.

Lines 98-104: Do authors mean “uninfected red blood cells and red blood cells infected with …”

What are the controls for this experiment?

Lines 105-113: Clarify the formula for “Phagocytic capacity and index”. What is being measured?

Explain the use of “Panotic kit”

Lines 114-121: Chemical formula should be written correctly. In line 117, 0.1% and in line 119, 550 nm wavelength. Provide additional information regarding the standard curve that was generated.

Line 123-132: Specify products generated from l-arginase instead of writing “some”. How were macrophages collected? Was the thermoblock used for heating rotatory or stationary? Values of 400 and 600 rpm are shown. Provide and explanation. This section needs to be revised for grammar and syntax. Spectrophotometer measurements at 550 nm wavelength.

Lines 134-135: Clarify groups analyzed.

Results

Figures 1-6: The labeling for the graphs are incomplete. On x-axis “CUR” is indicated on the right side of the graph in Figures 1-5. What is on the left side? Use a different shading for the bar graphs for easier differentiation of treatments.

Specify if controls or macrophage treatment with (+) or without (-) curcumin for internalization of zymosan, iRBC and RBC.

What stage of P. berghei was used? Was this a short term culture of the blood stage of P. berghei?

There are references to “infected macrophages”. Provide an explanation. Are the authors indicating that macrophages are infected with P. berghei?

Revise the grammar in the results section for clarity.

Line 247: Correct “Statistical analyzes…”

Discussion

Lines 260-349: Revise for clarity and correct for syntax, grammar and spelling.

Line 264: “…interaction with immune cells.”

Line 274: “…the average of phagocytized particles…”

Figures in the appendices also need to be labeled clearly.

Extensive revision of the manuscript is required to correct grammar, syntax and spelling.

Author Response

(The authors gave the same response as above.)

Round 2

Reviewer 1 Report

The quality of the manuscript has improved significantly after the revision.

Most of my concerns have been addressed. 

Only one point - the analysis of the mTOR/PI3K/Akt pathway - remained unprocessed. Of course, the difficulties that have arisen as a result of the pandemic are understandable for everyone. Nevertheless, I still consider this point to be essential in order to make the paper fit for publication in 'Pharmaceuticals'

Additionally, the licence to conduct the described animal experiments is not mentioned.

Author Response

Comments:

The quality of the manuscript has improved significantly after the revision.

Most of my concerns have been addressed.

We appreciate your reviews and comments. They were invaluable in improving the structure and quality of the article.

  1. Only one point - the analysis of the mTOR/PI3K/Akt pathway - remained unprocessed. Of course, the difficulties that have arisen as a result of the pandemic are understandable for everyone. Nevertheless, I still consider this point to be essential in order to make the paper fit for publication in 'Pharmaceuticals'

Thank you for your punctual observation. As mentioned previously, our purpose is to interpret and perform additional evaluations and experiments to improve our understanding of how curcumin could impact macrophage function (by modulating the mTOR/PI3K/Akt signaling pathway), thereby affecting their polarization following Plasmodium berghei infection. This point of discussion remains a line of research in our laboratory. Our goal is to obtain the necessary funding to acquire commercial kits and supplies to carry out new in vivo and in vitro experiments to answer these and other open questions. The discussion aims to analyze and describe the importance of our findings in light of what is already known about malaria infection. Nevertheless, we believe that our data about the impact of curcumin as a stabilizer of macrophage polarization in Plasmodium infection is already insightful information that fits the Pharmaceuticals Journal and will be further investigated.

  1. Additionally, the licence to conduct the described animal experiments is not mentioned.

Thank you for your insightful observation. This information was corrected in the text with the approval of the Animal Ethics Committee of the UFJF (reference number 019/2018).

I look forward to hearing from you soon.

Yours sincerely

Mara Celes

Reviewer 2 Report

YES, the authors followed point by point all the changes recommended in my first review report and provided a much-improved version of the manuscript.

I now consider that the article can be published in this last presented form.

The final decision will be made by the Academic Editor.

Thank you very much!

04 October 2023

Author Response

Comments:

YES, the authors followed point by point all the changes recommended in my first review report and provided a much-improved version of the manuscript.

I now consider that the article can be published in this last presented form.

The final decision will be made by the Academic Editor.

Thank you very much!

We appreciate the constructive reviews, suggestions, and well-made recommendations. They have provided us the opportunity to respond point-by-point and, as a result, improve the final version of the manuscript.

Yours sincerely

 Mara Celes

Reviewer 3 Report

The quality of the manuscript has improved after the revision, and authors addressed most of my concerns. 

Author Response

Comments:

The quality of the manuscript has improved after the revision, and authors addressed most of my concerns.

We appreciate the constructive reviews, suggestions, and well-made recommendations. They have provided us the opportunity to respond point-by-point and, as a result, improve the final version of the manuscript.

Best Regards,

Mara Celes

Reviewer 4 Report

Line 105: change the spelling of "dimethyl sulfoxide"

Line 117: clarify "zymosan A de Saccharomyces cerevisiae"

Line 121: CO2 (2 is subscript)

Line 135: 0.1%

Line 144: clarify "... used for heating rotary"

Lines 305-307: Dendritic cells (and macrophages) are not usual host cells for P. berghei infection.

Minor English corrections required.

Author Response

Comments:

Line 105: change the spelling of "dimethyl sulfoxide"

Line 117: clarify "zymosan A de Saccharomyces cerevisiae"

Line 121: CO2 (2 is subscript)

Line 135: 0.1%

Line 144: clarify "... used for heating rotary"

Lines 305-307: Dendritic cells (and macrophages) are not usual host cells for P. berghei infection.

Thank you for your insightful observation. All requested information has been corrected in the text and highlighted in green.

Comments on the Quality of English Language: Minor English corrections required.

Thanks for your comment and recommendations, the text has been revised by the MDPI English editing service (English editing ID: English-71245) and has undergone new revision for the authors.

Best Regards,

Mara Celes

Round 3

Reviewer 1 Report

the manuscript is now fit for publication in "Pharmaceutics"